mechanical engineering/computer modelling and simulation/acoustics

interior acoustics, coupled spaces, steady-state sound field, modal expansion method, sound damping, Green's function

**Author for correspondence:**
Mirosław Meissner
e-mail: mmeissn@ippt.pan.pl

# Investigation of damping effects on low-frequency steady-state acoustical behaviour of coupled spaces

Mirosław Meissner and Krzysztof Wiśniewski

Institute of Fundamental Technological Research, Polish Academy of Sciences, Pawińskiego 5B, 02–106, Warsaw, Poland

MM, 0000-0003-3885-0273; KW, 0000-0002-8835-0261

In the low-frequency range, the acoustical behaviour of enclosed spaces is strongly influenced by excited acoustic modes resulting in a spatial irregularity of a steady-state sound field. In the paper, this problem has been examined theoretically and numerically for a system of coupled spaces with complex-valued conditions on boundary surfaces. Using a modal expansion method, an analytic formula for Green's function was derived allowing to predict the interior sound field for a pure-tone excitation. To quantify the spatial irregularity of steady-state sound field, the parameter referred to as the mean spatial deviation was introduced. A numerical simulation was carried out for the system consisting of two coupled rectangular subspaces. Eigenfunctions and eigenfrequencies for this system were determined using the high-accuracy eigenvalue solver. As was evidenced by computational data, for small sound damping on absorptive walls the mean spatial deviation peaks at frequencies corresponding to eigenfrequencies of strongly localized modes. However, if the sound damping is much higher, the main cause of spatial irregularity of the interior sound field is the appearance of sharp valleys in a spatial distribution of a sound pressure level.

## 1. Introduction

The main objective of interior acoustics is to investigate the steady-state and transient acoustical behaviour of enclosed spaces. There are many theoretical methods for modelling the interior sound field and among them are statistical-acoustic methods, acoustic diffusion equation model, geometrical acoustics approaches, element-based techniques, wave-based method and modal expansion method. Statistical-acoustic methods [1] assume a uniform distribution of the acoustic energy in the field because they are based on the diffuse sound field hypothesis. The acoustic

**Figure 1.** Schematic view of considered coupled spaces and boundary problem definition.

diffusion equation model [2] is an extension of the statistical theory to spatially varying sound field. Geometrical acoustics techniques [3] are suitable for high sound frequencies, and the ray tracing method [4], the beam tracing method [5] and the image source method [6], also known as the mirror source method [7], are the most popular methods for geometric modelling. In contrast to geometrical methods, element-based approaches provide a complete description of the interior sound field because they solve the wave equation after suitable space discretization. The most common among these numerical techniques are the finite-element method (FEM) [8] and the boundary element method (BEM) [9]. Besides the FEM, BEM and all their variations, there is another family of methods, the so-called Trefftz methods which differ from the FEMs by the choice of shape and weighting functions. Examples of Trefftz-based methods are the wave-based method [10] and the method of fundamental solutions [11]. The finite-difference time-domain (FDTD) method is a numerical technique that simulates the time-dependent acoustic field using discrete approximations of the spatial derivative operators and an explicit time-stepping [12]. Alternatives to the FDTD technique are the pseudospectral time-domain method [13] and the adaptive rectangular decomposition, which achieve a good accuracy with a much coarser spatial discretization [14]. The modal expansion method (MEM) yields the acoustic modes of pressure vibrations inside enclosed spaces, and the sound field is expressed as a linear combination of these modes [15]. This method is more difficult to apply for irregularly shaped cavities [16] and coupled spaces [17], but it fully describe a wave nature of the sound field like a diffraction and a creation of standing waves. The MEM also enables to identify typical modal effects such as a modal degeneracy [18] and a localization of modes [19].

In this paper, the MEM is implemented to model a low-frequency steady-state acoustical behaviour of coupled spaces with complex-valued conditions on boundary surfaces. Using a modal expansion of a sound pressure, a general solution of the inhomogeneous wave equation is found which allowed us to determine Green's function. To quantify the spatial irregularity of steady-state sound field, the parameter called the mean spatial deviation is defined. A theoretical method is numerically tested for a system consisting of two coupled rectangular subspaces. Eigenfunctions and eigenfrequencies for this system are calculated using the FEAST eigenvalue solver. Based on calculation results, the influence of damping properties of sound absorbing walls on the mean spatial deviation is investigated, and changes in a steady-state distribution of a pressure amplitude are analysed. In the last part of the paper, major research findings of this study are summarized and concluding remarks are given.

## 2. Theoretical modelling

Coupled spaces in a of form the enclosed three-dimensional domain $\Omega$ filled by an air are considered (figure 1). Characteristic properties of the air are the speed of sound $c$ and the air density $\rho$. When inside $\Omega$ there is a volume sound source, spatial and temporal behaviours of a sound field are governed by the inhomogeneous wave equation

$$\left[\Delta - \frac{1}{c^2}\frac{\partial^2}{\partial t^2}\right]p(\mathbf{r}, t) = -q(\mathbf{r}, t), \tag{2.1}$$

where $\Delta = \partial^2/\partial x^2 + \partial^2/\partial y^2 + \partial^2/\partial z^2$ is the Laplace operator, $p$ is the sound pressure, $q$ is the volume source term and $\mathbf{r} = (x, y, z)$ is a position coordinate of a field point. The pressure $p$ must satisfy the causality condition. It also fulfils initial conditions defined for the time $t_0$. It is assumed that $p(\mathbf{r}, t_0)$ and $p'(\mathbf{r}, t_0) = \partial p(\mathbf{r}, t)/\partial t|_{t=t_0}$ are non-zero, thus, these conditions must appear in a general solution of the wave equation (2.1). A boundary of the domain $\Omega$ is denoted by $\Gamma$ and it is partitioned into two

parts: $\Gamma_r$, which is acoustically hard and $\Gamma_z$, where a sound absorptive material with the impedance $Z$ is placed. On these parts, the following boundary conditions for the pressure gradient are applied:

$$\mathbf{r} \in \Gamma_r: \nabla p \cdot \mathbf{n} = 0 \tag{2.2}$$

and

$$\mathbf{r} \in \Gamma_z: \nabla p \cdot \mathbf{n} = -\frac{1}{c\zeta}\frac{\partial p}{\partial t}, \tag{2.3}$$

where $\nabla = \mathbf{i}\partial/\partial x + \mathbf{j}\partial/\partial y + \mathbf{k}\partial/\partial z$ is the gradient vector operator, the dot is a scalar product, $\mathbf{n}$ is the outward normal vector and $\zeta = Z/\rho c$ is the normalized impedance of absorptive material. The impedance's real part $\zeta_r$ represents the normalized resistance, whereas its imaginary part $\zeta_i$ is referred to as the normalized reactance. Equation (2.1) together with the boundary conditions (2.2) and (2.3) describes a generation of a sound field inside the domain $\Omega$ when it is subjected to a volume sound source. Since the temporal variability of this source can be arbitrary, it is possible to predict both steady-state and transient behaviours of a sound field inside $\Omega$.

## 2.1. Determination of Green's function

In the low-frequency range, dimensions of the domain $\Omega$ are comparable with a length of sound wave; thus, the method which is most appropriate for determining the interior sound field is the modal expansion method. According to the MEM, the solution of the wave equation (2.1) can be expressed as a linear combination of the eigenfunctions $\Phi_m$

$$p(\mathbf{r}, t) = \sum_{m=1}^{\infty} p_m(t)\Phi_m(\mathbf{r}), \tag{2.4}$$

where $p_m$ are time-dependent modal amplitudes. It is assumed that eigenfunctions $\Phi_m$ are mutually orthogonal and are normalized in the volume $V$ of the domain $\Omega$ by the relation

$$\int_V \Phi_m \Phi_n \, dv = \delta_{mn}, \tag{2.5}$$

where $\delta_{mn}$ is the Kronecker delta function. Each eigenfunction $\Phi_m$ is related to the corresponding natural eigenfrequency $\omega_m$ through the equation

$$\Delta \Phi_m = -\left(\frac{\omega_m}{c}\right)^2 \Phi_m. \tag{2.6}$$

A method for determining the amplitude $p_m$ relies on suitable transformation of equation (2.1). First, multiply both sides of equation (2.1) by $\Phi_m$ and integrate over the volume $V$. This gives

$$\frac{1}{c^2}\int_V \Phi_m \frac{\partial^2 p}{\partial t^2}\, dv - \int_V \Phi_m \Delta p \, dv = \int_{V_s} q(\mathbf{r}, t)\Phi_m(\mathbf{r})\, dv, \tag{2.7}$$

where $V_s$ is the source volume. The application of equations (2.4) and (2.5) in the first volume integral in equation (2.7), and the utilization of equations (2.2), (2.3)–(2.6) and Green's theorem [20],

$$\int_V (p\, \Delta \Phi_m - \Phi_m \Delta p)dv = \int_S (p\,\nabla\, \Phi_m - \Phi_m \nabla p) \cdot \mathbf{n}\, ds, \tag{2.8}$$

in the second volume integral leads to the following equation for the modal amplitude $p_m$:

$$\frac{\partial^2 p_m}{\partial t^2} + 2\sum_{\substack{n \neq m \\ n=1}}^{\infty} \gamma_{mn}\frac{\partial p_n}{\partial t} + 2(r_m + \mathrm{j}\varphi_m)\frac{\partial p_m}{\partial t} + \omega_m^2 p_m + c^2 \int_S p\, \frac{\partial \Phi_m}{\partial n}\, ds$$
$$= c^2 \int_{V_s} q(\mathbf{r}, t)\Phi_m(\mathbf{r})\, dv. \tag{2.9}$$

In the above equation, $\gamma_{mn}$ are modal coupling factors expressed as

$$\gamma_{mn} = \frac{c}{2}\int_{S_z} (\zeta_r - \mathrm{j}\zeta_i)|\zeta|^{-2}\, \Phi_m \Phi_n\, ds, \tag{2.10}$$

and the quantities $r_m$ and $\varphi_m$ are modal coefficients determined by

$$r_m = \frac{c}{2}\int_{S_z} \zeta_r|\zeta|^{-2}\, \Phi_m^2\, ds, \quad \varphi_m = -\frac{c}{2}\int_{S_z} \zeta_i|\zeta|^{-2}\, \Phi_m^2\, ds, \tag{2.11}$$

where $|\zeta| = \sqrt{\zeta_r^2 + \zeta_i^2}$ is a magnitude of the normalized impedance of absorptive material, $S_z$ is a surface of the part $\Gamma_z$ of a domain boundary and $j = \sqrt{-1}$ is the imaginary unit. In the low-frequency range, typical absorptive materials are characterized by a small sound damping [21], thus it is possible to assume that $|\zeta|^{-1}$ is much smaller than unity. In this case, series components in equation (2.9) containing the coefficients $\gamma_{mn}$ can be omitted and it is possible to approximate the functions $\Phi_m$ by real-valued eigenfunctions predicted for acoustically hard boundary. Thus, boundary conditions $\partial \Phi_m / \partial n = 0$ are met, therefore, equation (2.9) simplifies to the uncoupled differential equation

$$\frac{\partial^2 p_m}{\partial t^2} + 2(r_m + j\varphi_m)\frac{\partial p_m}{\partial t} + \omega_m^2 p_m = c^2 \int_{V_s} q(\mathbf{r}, t)\Phi_m(\mathbf{r})\,\mathrm{d}v = s_m(t), \tag{2.12}$$

where $s_m(t)$ is a modal source function. Equation (2.12) was solved using the method of variation of parameters [22], and a general solution that includes initial conditions has the following form:

$$\begin{aligned}
p_m(t) = {}& p_m(t_0)\,\mathrm{e}^{-(r_m + j\varphi_m)(t-t_0)}\big\{\cos[\Omega_m(t-t_0)]\cosh[\vartheta_m(t-t_0)] \\
& - j\sin[\Omega_m(t-t_0)]\sinh[\vartheta_m(t-t_0)]\big\} \\
& + \frac{\mathrm{e}^{-(r_m + j\varphi_m)(t-t_0)}[(\Omega_m + j\vartheta_m)p_m(t_0) + p'_m(t_0)]}{\Omega_m + j\vartheta_m} \\
& \times \big\{\sin[\Omega_m(t-t_0)]\cosh[\vartheta_m(t-t_0)] + j\cos[\Omega_m(t-t_0)]\sinh[\vartheta_m(t-t_0)]\big\} \\
& + \frac{\mathrm{e}^{-(r_m + j\varphi_m)t}}{2(\vartheta_m - j\Omega_m)} \\
& \times \bigg[\mathrm{e}^{(\vartheta_m - j\Omega_m)t}\int_{t_0}^t s_m(\tau)\,\mathrm{e}^{[r_m - \vartheta_m + j(\varphi_m + \Omega_m)]\tau}\,\mathrm{d}\tau - \mathrm{e}^{-(\vartheta_m - j\Omega_m)t}\int_{t_0}^t s_m(\tau)\,\mathrm{e}^{[r_m + \vartheta_m + j(\varphi_m - \Omega_m)]\tau}\,\mathrm{d}\tau\bigg],
\end{aligned} \tag{2.13}$$

where the quantities $\Omega_m$ and $\vartheta_m$ are determined by the following equations:

$$\Omega_m = \sqrt{\frac{\alpha_m + \sqrt{\alpha_m^2 + \beta_m^2}}{2}} \quad \text{and} \quad \vartheta_m = \sqrt{\frac{-\alpha_m + \sqrt{\alpha_m^2 + \beta_m^2}}{2}}, \tag{2.14}$$

where $\alpha_m = \omega_m^2 - r_m^2 + \varphi_m^2$ and $\beta_m = -2r_m\varphi_m$. To calculate Green's function, it is assumed that a sound excitation has a form of a point source located at the position $\mathbf{r} = \mathbf{r}'$, which generates the Dirac pulse at the time $t = t_0$. This means that $q(\mathbf{r}, t) = \delta(\mathbf{r} - \mathbf{r}')\delta(t - t_0)$ in equation (2.12), thus, the modal source function $s_m(\tau)$ in equation (2.13) is as follows:

$$s_m(\tau) = c^2 \delta(\tau - t_0)\Phi_m(\mathbf{r}'). \tag{2.15}$$

An insertion of equation (2.15) in equation (2.13) enables to determine the function $p_m(t)$ for a temporal impulse. Since integrals in equation (2.13) have the lower limit corresponding to a peak of the delta function $\delta(t - t_0)$, the integration was carried out from $t_0^- = t_0 - \varepsilon$, where $\varepsilon$ is positive and arbitrarily small. In this case, the interval of integration includes the value at which the delta function peaks, but $p_m(t_0^-)$ and $p'_m(t_0^-)$ are equal to zero, because of a causality condition. Finally, inserting $p_m(t)$ in equation (2.4) the following formula for Green's function can be found:

$$\begin{aligned}
G(\mathbf{r}, t \,|\, \mathbf{r}', t_0) = {}& c^2 \sum_{m=1}^\infty \frac{\mathrm{e}^{-(r_m + j\varphi_m)(t-t_0)}}{\Omega_m^2 + \vartheta_m^2}\big\{\sin[\Omega_m(t-t_0)]\cosh[\vartheta_m(t-t_0)] \\
& + j\cos[\Omega_m(t-t_0)]\sinh[\vartheta_m(t-t_0)]\big\}(\Omega_m - j\vartheta_m)\,\Phi_m(\mathbf{r}')\Phi_m(\mathbf{r}).
\end{aligned} \tag{2.16}$$

The function $G$ and its time derivative $\partial G / \partial t$ are zero for $t < t_0$ because if an impulse occurs at $t_0$, no effects of the impulse should be present at the earlier time (a causality condition).

## 2.2. Quantification of spatial irregularity of steady-state sound field

By using Green's function $G$ it is possible to predict a sound field generated inside the domain $\Omega$ by any sound source because when the source function $q(\mathbf{r}, t)$ is known, the pressure response to this excitation is described by the following equation [23]:

$$p(\mathbf{r}, t) = \int_{V_s} \int_{-\infty}^t q(\mathbf{r}', \tau)\,G(\mathbf{r}, t \,|\, \mathbf{r}', \tau)\,\mathrm{d}\tau\,\mathrm{d}v'. \tag{2.17}$$

The steady-state pressure response to a point source can be found assuming that in equation (2.17) the source function $q$ takes the form $q(\mathbf{r}', \tau) = Q\delta(\mathbf{r}' - \mathbf{r}_0)\,e^{j\omega\tau}$, where $\omega$ is the angular source frequency, $\mathbf{r}_0 = (x_0, y_0, z_0)$ determines a source position and the amplitude $Q$ is dependent on the source power $W$ according to the formula $Q = \sqrt{8\pi\rho c W}$ [24], where, as before, $\rho$ denotes the air density. Thus, after performing the volume and time integrations in equation (2.17), a formula for the steady-state pressure amplitude $P_c$ is found as

$$P_c(\mathbf{r}) = \sum_{m=1}^{\infty} (a_m + jb_m)\,\Phi_m(\mathbf{r}), \tag{2.18}$$

where the quantities $a_m$ and $b_m$ are determined by

$$a_m = \frac{Qc^2[r_m^2 + \Omega_m^2 - \vartheta_m^2 - (\omega + \varphi_m)^2]\Phi_m(\mathbf{r}_0)}{[(r_m + \vartheta_m)^2 + (\omega + \varphi_m - \Omega_m)^2][(r_m - \vartheta_m)^2 + (\omega + \varphi_m + \Omega_m)^2]} \tag{2.19}$$

and

$$b_m = -\frac{2Qc^2[r_m(\omega + \varphi_m) + \Omega_m\vartheta_m]\Phi_m(\mathbf{r}_0)}{[(r_m + \vartheta_m)^2 + (\omega + \varphi_m - \Omega_m)^2][(r_m - \vartheta_m)^2 + (\omega + \varphi_m + \Omega_m)^2]}. \tag{2.20}$$

Since the amplitude $P_c$ is complex, a quantity suitable for a prediction of the steady-state pressure response is the real pressure amplitude $P$ determined by absolute value of $P_c$, i.e.

$$P(\mathbf{r}) = \sqrt{P_c(\mathbf{r})P_c^*(\mathbf{r})}, \tag{2.21}$$

where an asterisk in a superscript denotes the complex conjugate. Thus, after inserting equation (2.18) into equation (2.21) one finds the following formula for the pressure amplitude:

$$P(\mathbf{r}) = \left\{ \left[\sum_{m=1}^{\infty} a_m \Phi_m(\mathbf{r})\right]^2 + \left[\sum_{m=1}^{\infty} b_m \Phi_m(\mathbf{r})\right]^2 \right\}^{1/2}. \tag{2.22}$$

As it results from equations (2.18) to (2.20), the amplitude $P$ is dependent on the source position $\mathbf{r}_0$ and the source frequency $\omega$ and, through the quantities $\Omega_m$, $\vartheta_m$, $r_m$ and $\varphi_m$, on the natural eigenfrequency $\omega_m$ as well as the real and imaginary parts of the impedance $\zeta$. Thus, for constant $\mathbf{r}_0$ and given source frequency $\omega$, equation (2.22) enables one to predict a spatial irregularity of the steady-state pressure amplitude for different values of $\zeta_r$ and $\zeta_i$.

A quantity more accurate for assessing a spatial irregularity of the sound field is the pressure level given by

$$L(\mathbf{r}) = 20\log\left[\frac{P(\mathbf{r})}{p_0}\right], \tag{2.23}$$

where $p_0 = 20\,\mu\text{Pa}$, because a knowledge of a spatial distribution of $L$ allows to evaluate how much its value deviates from point to point on the observation plane. Small variations in $L$ imply great steady-state response homogeneity, while large variations indicate high irregularity of this response. To quantify sound pressure level variation on the whole observation area, the parameter $D$ named the mean spatial deviation will be defined as

$$D = \left\{\frac{1}{S_p}\int_{S_p} [L(\mathbf{r}_p) - L_{av}]^2 ds\right\}^{1/2}, \tag{2.24}$$

where $S_p$ is the size of the observation area, $\mathbf{r}_p$ is a position coordinate on this area and $L_{av}$ is the average sound pressure level determined by

$$L_{av} = 20\log\left(\frac{P_{av}}{p_0}\right), \quad P_{av} = \frac{1}{S_p}\int_{S_p} P(\mathbf{r}_p)\,ds, \tag{2.25}$$

where $P_{av}$ is the average pressure amplitude on the observation area. The parameter $D$ is quantified in decibels and a value of $D$ close to zero dB means that the sound pressure field is uniform. If the parameter $D$ does not meet this requirement, there is a spatial irregularity of the sound pressure field.

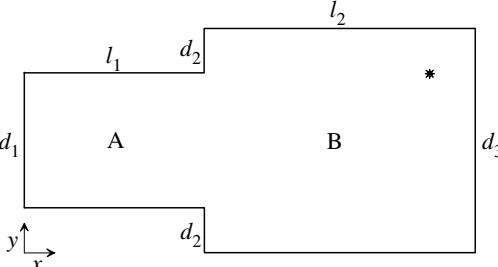

**Figure 2.** Horizontal plan view of examined system of coupled spaces consisting of two connected rectangular subspaces A and B. Symbol shows position of point sound source.

# 3. Description of examined system of coupled spaces

The objective of a numerical study was to simulate a steady-state sound field in a system of coupled spaces consisting of two connected cuboidal subspaces of the same height. This was motivated by the fact that such a configuration of coupled spaces can be found in many buildings and constructions. It was assumed that an air filling the coupled spaces is characterized by the speed of sound $c = 343 \text{ m s}^{-1}$ and the density $\rho = 1.21 \text{ kg m}^{-3}$. A horizontal cross-section of the examined system together with the associated coordinate system are sketched in figure 2. Simulations were run for the following dimensions of the system: $d_1 = 3$ m, $d_2 = 1$ m, $d_3 = 5$ m, $l_1 = 4$ m, $l_2 = 6$ m and $h = 3$ m, which is the height of subspaces. As may be noted, these dimensions seem realistic for coupled spaces encountered in practice. It was assumed that the system was excited by a point source with the power $W$ of $10^{-3}$ W located at the position: $x_0 = 9$ m, $y_0 = 4$ m, $z_0 = 1$ m.

In the tested system of coupled spaces, the bottom wall was assumed to be nearly hard acoustically, which means that a magnitude of wall impedance is very large but finite.[1] Sound damping in this system is provided by an absorptive material with the impedance $\zeta$ which is uniformly distributed on the side walls and the top wall. Absorbing properties of the material are described by the random-incident absorption coefficient $\alpha$ which is related to real and imaginary parts of the impedance $\zeta$ by the following expression [25]:

$$\alpha = \frac{8\zeta_r}{|\zeta|^2}\left[1 - \frac{\zeta_r \ln(1 + 2\zeta_r + |\zeta|^2)}{|\zeta|^2} + \frac{\zeta_r^2 - \zeta_i^2}{\zeta_i|\zeta|^2}\arctan\left(\frac{\zeta_i}{1 + \zeta_r}\right)\right]. \tag{3.1}$$

In figure 3, this relation is represented graphically for the absorption coefficient $0.1 \leq \alpha < 0.37$. The diagram shows contours of constant value of $\alpha$ in the complex $\zeta$-plane, i.e. abscissa and ordinate in this figure are the real and imaginary part of the impedance $\zeta$, respectively. If values of $\alpha$ are small, the expression in square brackets on the right-hand side of (3.1) approaches unity. This makes it possible to approximate equation (3.1) by the formula

$$(\zeta_r - R)^2 + \zeta_i^2 = R^2, \tag{3.2}$$

where $R = 4/\alpha$; therefore, contours of constant value of $\alpha$ represent circles of radius $R$ with the centre located at the point $\zeta_r = R$, $\zeta_i = 0$ (figure 3). From equation (3.1), it also follows that the necessary condition for sound damping is that the resistance $\zeta_r$ has a positive value. By contrast, there are no restrictions on the value of the reactance $\zeta_i$ because it can be positive or negative or zero. In the latter case, equation (3.1) can be transformed into the following form:

$$\alpha = \frac{8}{\zeta_r}\left[1 - \frac{2}{\zeta_r}\ln(1 + \zeta_r) + \frac{1}{1 + \zeta_r}\right]. \tag{3.3}$$

In the numerical study, the damping properties of the absorptive material were modelled as follows: it was assumed that the resistance $\zeta_r$ has a constant value, namely $\zeta_r = 15$, so changes in the damping properties were simulated by variations in the reactance $\zeta_i$. With these assumptions, the condition $|\zeta|^{-1} \leq 0.0667$ is obtained because the maximum value of $|\zeta|^{-1}$ is reached when $\zeta_r = 15$ and $\zeta_i = 0$. This maximum value is sufficiently small to consider the analysed system of coupled spaces as slightly damped. Using equation (3.3), it can be found that for the resistance $\zeta_r$ of 15, the absorption

---

[1]When a wall is acoustically hard, a magnitude of wall impedance is assumed to be theoretically infinite.

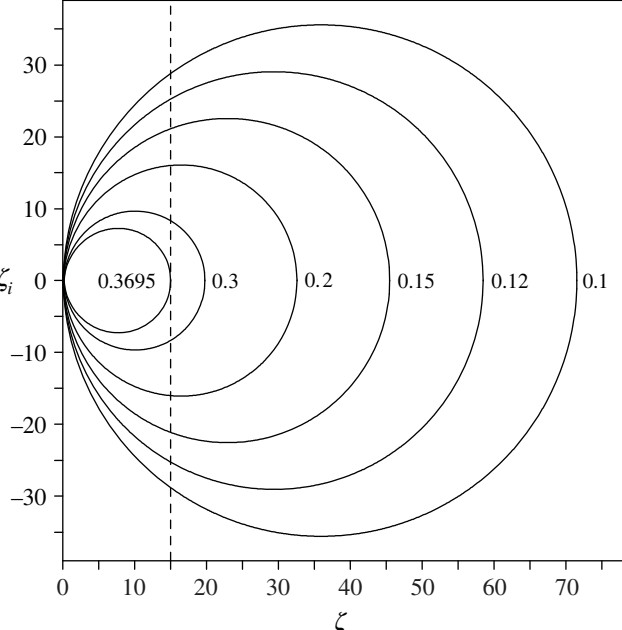

**Figure 3.** Contours of constant values of the absorption coefficient $\alpha$ in complex $\zeta$-plane.

coefficient amounts to 0.3695 and this value corresponds to $\alpha_{\max}$, i.e. the maximum value of $\alpha$ for $\zeta_r$ equal to 15. This result has a simple geometrical interpretation; namely, the straight line described in the complex $\zeta$-plane by $\zeta_r = 15$ represents the tangent line to the contour of constant value of the absorption coefficient when $\alpha$ is equal to $\alpha_{\max}$ (figure 3).

# 4. Simulation results and discussion

## 4.1. Eigenfunctions

As proved by the theoretical analysis, the steady-state sound fields depend on the eigenfunctions $\Phi_m$. For a considered system geometry, there is no analytical form of $\Phi_m$, thus, a determination of their form requires an application of numerical methods. However, the fact that an absorptive material provides a small sound damping enables one to assume that $\Phi_m$ are well approximated by real-valued eigenfunctions predicted for the acoustically hard boundary. In this case, the eigenfunctions $\Phi_m$ are replaced by the double-indexed eigenfunctions $\Phi_{\kappa\nu}$ whose dependence on the coordinate $z$ describes clearly defined cosine functions. Thus, the expression for the functions $\Phi_{\kappa\nu}$ can be written as

$$\Phi_{\kappa\nu}(\mathbf{r}) = \begin{cases} \sqrt{\frac{1}{h}}\,\Psi_\nu(x, y), & \kappa = 0,\ \nu > 0, \\ \sqrt{\frac{2}{V}}\,\cos(\pi\kappa z/h), & \kappa > 0,\ \nu = 0, \\ \sqrt{\frac{2}{h}}\,\cos(\pi\kappa z/h)\,\Psi_\nu(x, y), & \kappa > 0,\ \nu > 0, \end{cases} \tag{4.1}$$

where $\kappa$ and $\nu$ are non-negative integers and the eigenfunctions $\Psi_\nu$ are normalized over a horizontal cross-section of the system. The functions $\Psi_\nu$ and corresponding eigenfrequencies $\omega_\nu$ were computed in a two-dimensional mesh with 105 902 nodes (a distance between adjacent nodes amounted to 2 cm) using the FEAST eigenvalue solver which exhibits high accuracy and computational efficiency [26]. Since $\Psi_\nu = \sqrt{1/S_0}$ for $\nu = 0$, where $S_0$ is a surface of a horizontal cross-section of the system, the eigenfrequency $\omega_\nu$ corresponding to this eigenfunction is equal to zero. Finally, the eigenfrequencies $\omega_{\kappa\nu}$ of the examined system of coupled spaces were calculated from the following expression:

$$\omega_{\kappa\nu} = \sqrt{\left(\frac{\pi\kappa c}{h}\right)^2 + \omega_\nu^2}, \tag{4.2}$$

where the indices $\kappa$ and $\nu$ are not simultaneously equal to zero. By applying the FEAST, the eigenfunctions $\Phi_{\kappa\nu}$ for the first 500 room modes were calculated and the eigenfrequencies $f_{\kappa\nu} = \omega_{\kappa\nu}/2\pi$

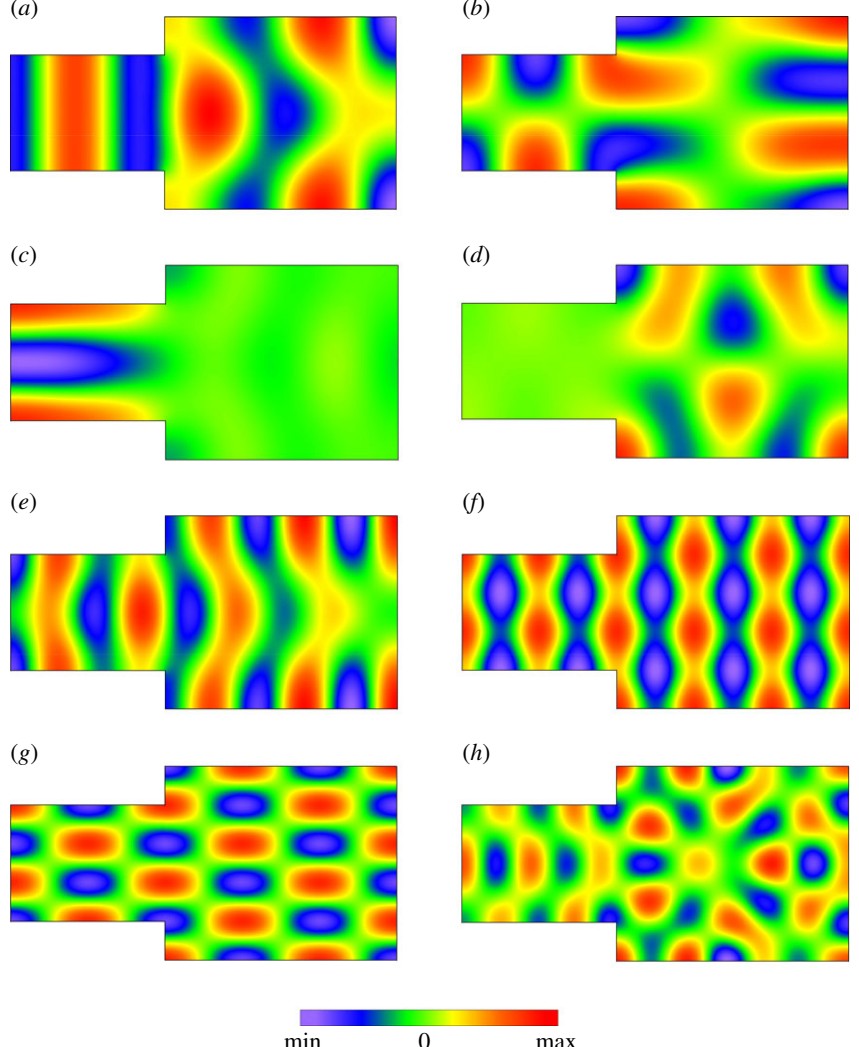

**Figure 4.** Shapes of the eigenfunctions $\Psi_\nu$ for the mode number $\nu$: (a) 15, (b) 17, (c) 20, (d) 22, (e) 33, (f) 39, (g) 49, (h) 62. Frequencies $f_\nu = \omega_\nu / 2\pi$ for these functions are (in Hz): (a) 102.13, (b) 106.25, (c) 115.99, (d) 119.11, (e) 151.39, (f) 171.53, (g) 191.77, (h) 216.01.

corresponding to these functions were from the range 17.5–308.5 Hz. Examples of computed shapes of the functions $\Psi_\nu$ are plotted in figure 4 in the form of filled contour maps which are a two-dimensional representation of three-dimensional data. These results imply that for some modes the acoustic energy can be concentrated inside the one of subspaces (figure 4c,d). This effect is called the localization of modes and is characteristic for irregularly shaped spaces because in a cuboidal space all eigenmodes are delocalized. As previously assumed, the eigenfunctions $\Phi_{\kappa\nu}$ are normalized in the system volume $V$, thus, the integral of $\Phi_{\kappa\nu}^2$ over $V$ equals unity. In order to identify the localized model, one should then compute the non-dimensional parameter

$$l_{\kappa\nu} = \int_{V_A} \Phi_{\kappa\nu}^2 \, dv, \tag{4.3}$$

where $\kappa \geq 0$, $\nu > 0$ and $V_A$ is the volume of the subspace A. Therefore, a mode is localized in the subspace A when the parameter $l_{\kappa\nu}$ is close to unity and it is localized in the subspace B when the value of $l_{\kappa\nu}$ is close to zero. In the numerical study, it was assumed that an eigenmode is recognized as a strongly localized mode if $l_{\kappa\nu}$ or $1 - l_{\kappa\nu}$ has value above 0.95.

It is worth mentioning that among computed eigenmodes, there are also modes which are totally delocalized. It follows from the fact that they represent eigenmodes of the cuboidal space having the height of 3 m and the width of 5 m, as the system under study, and the length of 10 m being a sum of

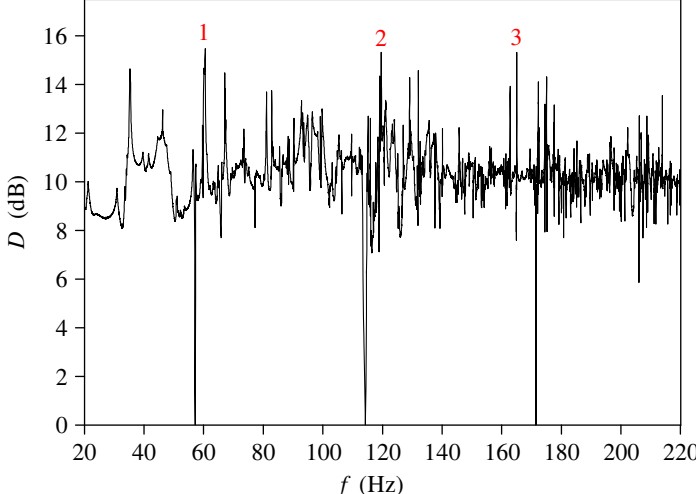

**Figure 5.** Frequency dependence of the mean spatial deviation $D$ for the absorption coefficient $\alpha$ of $10^{-4}$. Numbered peaks occur at frequencies: 60.51, 119.6 and 165.03 Hz.

$l_1$ and $l_2$ (figure 2). Exemplary shapes of the eigenfunctions $\Psi_\nu$ corresponding to these modes are shown in figure 4$f$,$g$.

## 4.2. Mean spatial deviation

An irregular distribution of the steady-state pressure amplitude $P$ inside coupled spaces originates from a strong dependence of this amplitude on a spatial distribution of the eigenfunctions $\Phi_{\kappa\nu}$. For a small sound damping on wall surfaces, this leads to highly position-sensitive acoustic responses which result in a spatial variability of the sound field. To quantify this variability, the mean spatial deviation $D$ was defined (equation (2.24)) and the purpose of a numerical simulation is to determine a frequency dependence of $D$ for different values of the absorption coefficient $\alpha$. This will enable us to identify sound frequencies for which there is a high irregularity of the sound field and to investigate how this irregularity is influenced by absorption characteristics of walls. Calculations were carried out for the observation plane located at the distance $z = 1.6$ m from the high reflecting bottom wall. As stated in §3, sound damping inside coupled spaces was provided by an absorptive material with the complex impedance $\zeta = \zeta_r + j\zeta_i$. This material was uniformly distributed on the side walls and the top wall. Since the absorption coefficient $\alpha$ of the material was chosen as input data in a numerical implementation, the reactance $\zeta_i$ was determined numerically from equation (3.1) assuming constant value of the resistance $\zeta_r$. As shown in figure 3, when the reactance $\zeta_i$ is non-zero, for a constant value of $\zeta_r$ there are two values of $\zeta_i$ that have the same absolute value but differ in signs. In this study, calculations were made assuming that the reactance $\zeta_i$ is non-positive.

In the first stage of a numerical simulation, the case of negligible sound damping was considered, assuming that all walls are nearly hard acoustically. It was established that the absorption coefficient $\alpha$ corresponding to this case has the value $10^{-4}$, and it is equivalent to the wall impedance $\zeta = 15 - 1094j$. The aim of numerical test was to identify main causes of spatial irregularity of steady-state sound pressure when sound damping is negligibly small. Simulation results gained in this case are presented in figure 5. Numerical data depict changes in the mean spatial deviation $D$ with the source frequency in the band 20–220 Hz. To accurately reconstruct a frequency dependence of $D$, in calculations the frequency step of 0.01 Hz was applied. In numerical predictions, all calculated modes of coupled spaces were used for determining a sound pressure field, to allow the residues from modes in the region 220–308.5 Hz to influence the frequency response below 220 Hz.

As shown in figure 5, in a frequency dependence of $D$ there are many intense peaks apparent for some frequencies. The most intense peaks denoted in figure 5 by numbers 1, 2 and 3 occur at frequencies: 60.51, 119.6 and 165.03 Hz and reach values above 15 dB. This proves that in these cases there is a very large irregularity of the sound field. This finding is confirmed by graphs in figure 6 showing mapped distributions of a sound pressure amplitude $P$ on the observation plane for frequencies of these peaks. An analysis of simulation data has demonstrated that intense peaks of $D$ are a direct result of the effect of modal localization, because frequencies of these peaks are in

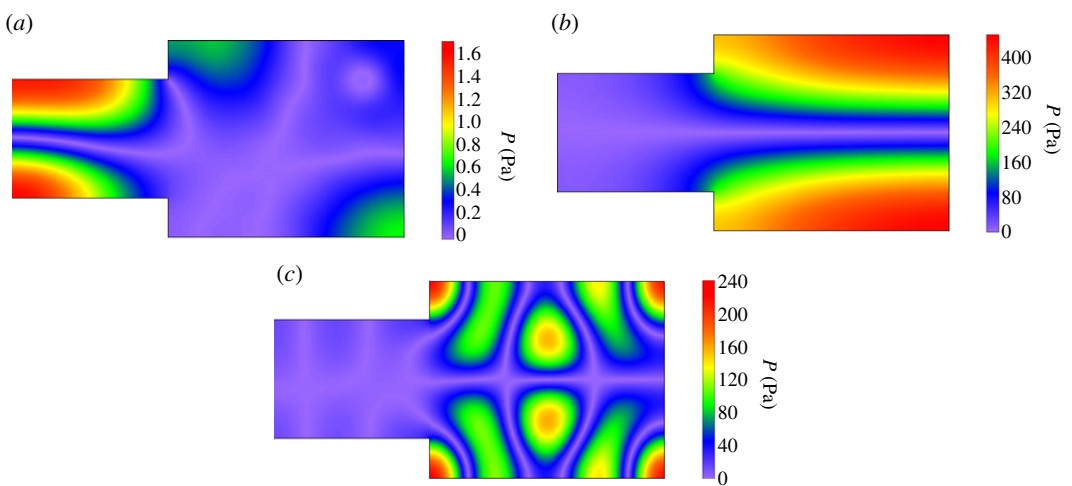

**Figure 6.** Mapped distribution of the sound pressure amplitude $P$ on the observation plane $z = 1.6$ m for source frequencies: (a) 60.51 Hz, (b) 119.6 Hz, (c) 165.03 Hz. Absorption coefficient $\alpha$ of $10^{-4}$.

**Table 1.** Frequencies $f_{\kappa v}$ of strongly localized modes in the frequency band 20–220 Hz together with corresponding mode numbers $\kappa$ and $v$.

| $\kappa$ | $v$ | $f_{\kappa v}$ (Hz) | $\kappa$ | $v$ | $f_{\kappa v}$ (Hz) | $\kappa$ | $v$ | $f_{\kappa v}$ (Hz) |
|---|---|---|---|---|---|---|---|---|
| 0 | 3 | 35.31 | 2 | 3 | 119.66 | 3 | 3 | 175.10 |
| 0 | 4 | 46.27 | 2 | 4 | 123.34 | 3 | 4 | 177.63 |
| 0 | 6 | 60.08 | 2 | 6 | 129.16 | 2 | 27 | 179.00 |
| 1 | 3 | 67.19 | 1 | 20 | 129.31 | 3 | 6 | 181.72 |
| 0 | 9 | 69.81 | 1 | 22 | 132.11 | 3 | 9 | 185.16 |
| 1 | 4 | 73.55 | 2 | 9 | 133.96 | 0 | 56 | 206.20 |
| 1 | 6 | 82.93 | 0 | 27 | 137.72 | 3 | 20 | 207.04 |
| 1 | 9 | 90.22 | 1 | 27 | 149.12 | 3 | 22 | 208.80 |
| 0 | 20 | 115.99 | 2 | 20 | 162.87 | 1 | 56 | 213.97 |
| 0 | 22 | 119.11 | 2 | 22 | 165.10 | 3 | 27 | 219.95 |

agreement with frequencies of strongly localized modes (table 1). For example, using computational data from table 1, it is easy to check that frequencies of the most intense peaks correspond to the frequencies of strongly localized modes having the following mode numbers: $\kappa = 0$ and $v = 6$, $\kappa = 2$ and $v = 3$, $\kappa = 2$ and $v = 22$. The effect of modal localization is characteristic for coupled spaces because among 200 modes found in the frequency range 20–220 Hz, 30 modes were recognized as strongly localized modes (table 1). This means that in the considered frequency band, the strongly localized modes account for 15% of all modes.

As may be seen in figure 5, the value of $D$ in the considered frequency band is above 7 dB with the exception of three frequencies for which there are sharp minima in $D$. This fact has a simple explanation; namely, at these frequencies values of $D$ are close to zero because they correspond to frequencies of $z$-axial modes for which the eigenfunctions $\Phi_{\kappa v}$ are not dependent on the coordinates $x$ and $y$. Frequencies of $z$-axial modes can be calculated from equation (4.2) assuming $\omega_v = 0$ and then inserting $\kappa = 1, 2, 3\ldots$

The way to reduce intense peaks of $D$ is to increase the sound attenuation inside the coupled spaces. This is due to the fact that with increased sound damping, the energy of strongly localized mode is physically attenuated. Consequently, neighbouring modes have a much greater impact on a distribution of a sound field for a frequency of localized mode, resulting in a reduction of point-to point variations in a sound pressure level. In the tested system of coupled spaces the sound damping is provided by an absorptive material uniformly distributed on the side walls and the top wall.

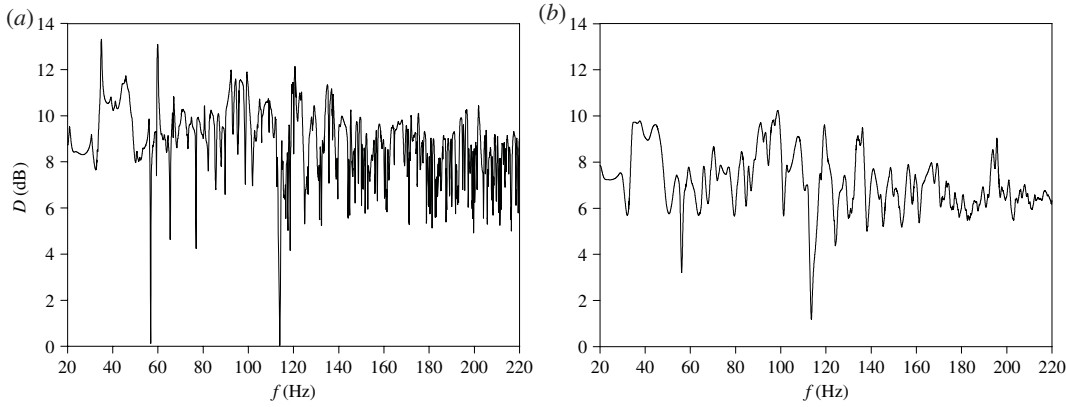

**Figure 7.** Frequency dependence of the mean spatial deviation $D$ for the absorption coefficient $\alpha$ equal to: (a) 0.01, (b) 0.1.

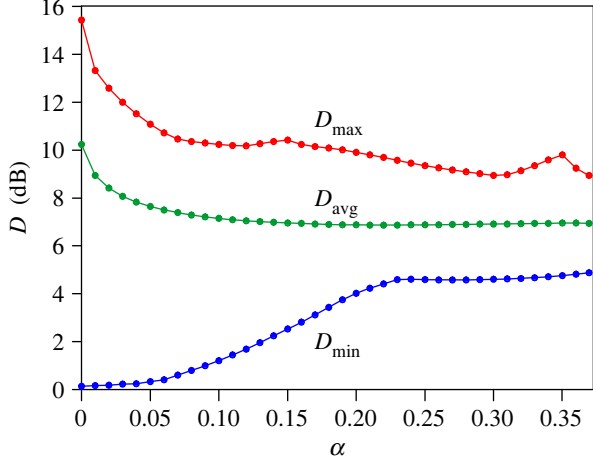

**Figure 8.** Changes in $D_{max}$, $D_{min}$ and $D_{avg}$ with the absorption coefficient $\alpha$. Coloured dots indicate calculation results.

As assumed in §3, damping properties of this material are described by the absorption coefficient $\alpha$ which is dependent on the impedance $\zeta$ according to equation (3.1). Calculation results in figure 7 show frequency dependence of the mean spatial deviation $D$ for the absorption coefficient $\alpha$ equal to 0.01 and 0.1. These values of $\alpha$ correspond to the impedance $\zeta$ of $15 - 107.1j$ and $15 - 28.8j$, respectively. A comparison of figures 5 and 7a proves that the increase in $\alpha$ to the value of 0.01 results in a reduction of most intense peaks of $D$ occurring at frequencies of strongly localized modes. However, the sharp peaks are still visible for frequencies close to 35 and 60 Hz, which, as shown by the data in table 1, correspond almost exactly to frequencies of the first and third strongly localized mode. It should also be emphasized that a minimum value of $D$ is still close to zero because the increase of $\alpha$ to 0.01 does not eliminate strong drops of $D$ occurring at frequencies of z-axial modes. A more effective reduction of intense peaks and strong drops in the frequency dependence of $D$ takes place when the absorption coefficient $\alpha$ grows to 0.1 (figure 7b). However, even for such considerable increase in the sound damping, the maximum value of $D$ is still greater than 10 dB, whereas the minimum value is below 2 dB.

To determine the effect of the sound damping on the maximum and minimum values of $D$, in figure 8 changes in $D_{max}$ and $D_{min}$ on the coefficient $\alpha$ are depicted in the whole range of its possible values, i.e. from $10^{-4}$ to $\alpha_{max} = 0.3695$. Additionally, in figure 8 changes in $D_{avg}$, i.e. the average value of $D$ in the considered frequency range, are also shown. Calculations were carried out with an increment of $\alpha$ of 0.01 and the obtained results are indicated by coloured dots. As evidenced in figure 8, the rapid decrease in $D_{max}$ occurs up to the coefficient $\alpha$ of 0.07, and this behaviour is caused by a substantial reduction in the peaks of $D$ appearing at frequencies of strongly localized modes. For values of $\alpha > 0.07$, variations in $D_{max}$ are much smaller because its values range from 8.94 ($\alpha = 0.3$) to 10.41 ($\alpha = 0.15$). On the other hand, the increase of $D_{min}$ visible up to $\alpha$ of 0.23 is associated with a disappearance of strong drops of $D$ occurring at frequencies of z-axial modes. The important finding

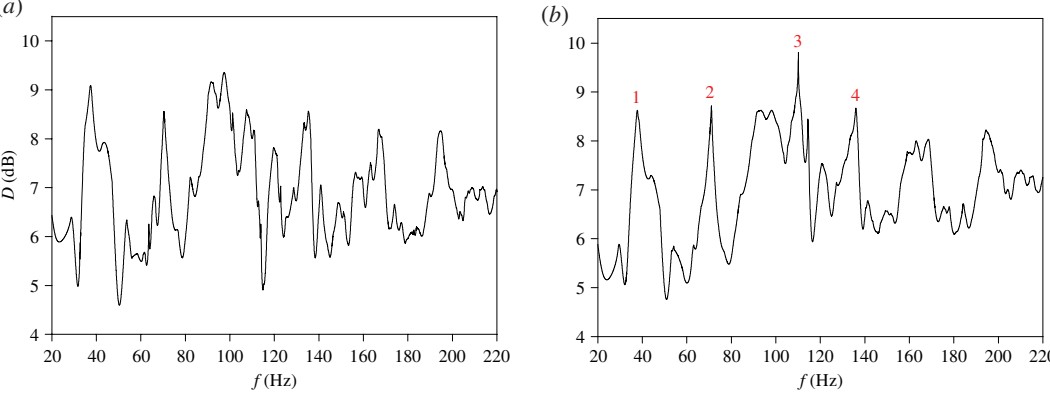

**Figure 9.** Frequency dependence of the mean spatial deviation *D* for the absorption coefficient $\alpha$ equal to: (*a*) 0.25, (*b*) 0.35. Numbered peaks occur at frequencies: 37.8, 71.6, 110.2 and 136 Hz.

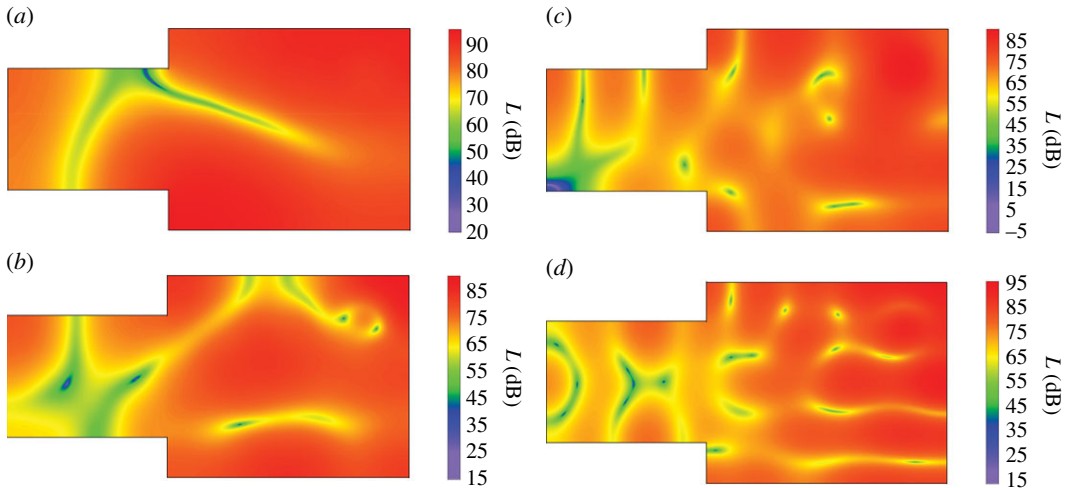

**Figure 10.** Mapped distribution of the sound pressure level *L* on the observation plane $z = 1.6$ m for source frequencies: (*a*) 37.8 Hz, (*b*) 71.6 Hz, (*c*) 110.2 Hz, (*d*) 136 Hz. Absorption coefficient $\alpha$ equal to 0.35.

resulting from figure 8 is such that for $\alpha$ changing from 0.25 to $\alpha_{max}$, a sound damping inside coupled spaces has a negligible effect on $D_{min}$ and $D_{avg}$. Variations in $D_{max}$ are much more pronounced because $D_{max}$ reaches a local maximum for the coefficient $\alpha$ of 0.35. As can be inferred, this is due to the appearance of a sharp peak in a frequency dependence of $D$ because the increase in $D_{max}$ is not accompanied by an increase in $D_{avg}$. This deduction is confirmed by graphs in figure 9 illustrating a frequency dependence of the mean spatial deviation $D$ for the absorption coefficient $\alpha$ of 0.25 and 0.35. These values of $\alpha$ correspond to the impedance $\zeta$ equal to $15 - 12j$ and $15 - 4.1j$, respectively. A comparison between figure 9*a,b* leads to the conclusion that the increase in $\alpha$ from 0.25 to 0.35 does not entail big changes in the frequency dependence of $D$, and the only significant difference is a sharp peak of $D$ at the frequency of 110.2 Hz occurring for $\alpha$ of 0.35.

In figure 9*b*, the most intense peaks are numbered from 1 to 4. In these peaks, the mean spatial deviation $D$ reaches values above 8.5 dB, which indicates a considerable irregularity of a sound field. To find the cause of this irregularity, for frequencies of these peaks a distribution of the sound pressure level $L$ was simulated and the results obtained are presented in figure 10. These data show that the substantial irregularity of a sound field is due to a presence of sharp valleys in a distribution of the pressure level $L$. In the greatest valley depth, the minimum value of $L$ is reached and for the cases depicted in figure 10*a,b,d*, this value is from the range 15–21 dB. As seen in figure 10*c*, the drop in the pressure level $L$ is dramatically large for the frequency of 110.2 Hz because in this case the minimum value of $L$ amounts to −3.5 dB.

# 5. Conclusion

In this paper, the impact of damping effects on low-frequency steady-state acoustical behaviour of coupled spaces has been examined. To determine a sound field for a pure-tone excitation, the analytic form of Green's function for an enclosed space with complex-valued boundary conditions was derived using the modal expansion method. The new parameter referred to as the mean spatial deviation was introduced to quantify the irregularity of steady-state sound field. The numerical study was performed for a system consisting of two coupled rectangular subspaces which is often encountered in practice. To ensure high accuracy in the calculation of eigenfunctions and eigenfrequencies, the FEAST eigenvalue solver was applied. Simulation results have demonstrated that intense peaks in a frequency dependence of the mean spatial deviation are a direct result of the modal localization. This effect is characteristic for irregularly shaped or coupled spaces and manifests itself through a high concentration of acoustic energy in a small space. As was evidenced by numerical data, strongly localized modes account for about 15% of all modes in the considered frequency range. As expected, the increase of sound damping contributes to a reduction of the most intense peaks resulting from the modal localization. However, it turned out that after a substantial growth in the sound damping, the mean spatial deviation does not reduce to zero due to the appearance of sharp valleys in a distribution of a sound pressure level. The mechanism of formation of these valleys is not well recognized, therefore further research on this subject is needed.

A spatial irregularity of a sound field occurs in small rooms because at low frequencies room acoustic quality is strongly influenced by excited room modes. This irregularity can give rise to highly position-sensitive acoustical responses that significantly limit a correct perception of speech and music. Therefore, the proposed theoretical method can be applied in the design or acoustic treatment of small rooms such as performance studios, studio control rooms, listening rooms, audio programme assessment rooms and small conference and lecture rooms where speech, music or listening is part of normal use.

Data accessibility. Eigenfunctions and eigenfrequencies calculated during this study are available via the Dryad Digital Repository: https://doi.org/10.5061/dryad.bzkh1895s [27].
Authors' contributions. M.M. conceived the original idea for this study, contributed to the theoretical development and wrote the manuscript; K.W. created numerical implementations and participated in the analysis of computational data. Both authors gave final approval for publication.
Competing interests. We declare we have no competing interests.
Funding. This work was financially supported by the National Science Center (NCN), Poland under the project 'Steady-state and transient energetic sound field parameters for objective evaluation of acoustics of enclosed spaces: theoretical modelling and computer simulations', grant agreement no. 2016/21/B/ST8/02427.

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
