## [Reviewer comments · Royal Society Open Science]

Review History

RSOS-200514.R0 (Original submission)

Review form: Reviewer 1

Is the manuscript scientifically sound in its present form?

Yes

Are the interpretations and conclusions justified by the results?

Yes

Is the language acceptable?

Yes

Do you have any ethical concerns with this paper?

No

Have you any concerns about statistical analyses in this paper?

No

Recommendation?

Accept with minor revision (please list in comments)

Comments to the Author(s)

The authors have stated that "The way to reduce the value of the parameter D is to increase the sound attenuation inside the coupled spaces. This is due to the fact that with increased sound damping, the energy of the acoustic modes is physically attenuated, reducing the point-to-point variations in the a sound pressure level." This explanation is not correct. Because the authors' standard deviation is calculated using decibel values, changing the modal energy will not change D. The reduction in D with increasing sound absorption is due to the increase in modal bandwidth which increases the modal overlap. The increased modal overlap means that more modes are excited at a given frequency and this reduces the spatial standard deviation of the sound pressure level in the room (D). The increase of modal bandwidth with increasing sound absorption also explains why the variation of D with frequency becomes slower as the sound absorption is increased.

The authors' use of a sound source position that is equidistant (1 m) from the floor and two of walls will create an unusual situation. The authors should repeat their calculations with the distances of the source chosen so that they are not in the ratio of small integers to each other.

Review form: Reviewer 2

Is the manuscript scientifically sound in its present form?

Yes

Are the interpretations and conclusions justified by the results?

Yes

Is the language acceptable?

Yes

Do you have any ethical concerns with this paper?

No

Have you any concerns about statistical analyses in this paper?

No

Recommendation?

Accept with minor revision (please list in comments)

Comments to the Author(s)

This is a nicely presented paper. Well done! I have four minor comments:

1. I would suggest including some practical reasons why spatial variation of pressure fields may be an important metric in the design of acoustic enclosures.
2. The authors should check their use of "specific acoustic impedance" to describe their parameter zeta. Specific acoustic impedance is defined as pressure/velocity and is given by $Z = \rho \cdot c \cdot \zeta$. Zeta is simply a convenient non-dimensional ratio of specific acoustic impedance to the characteristic impedance of the fluid.
3. There is a English usage error on line 43 of page 3.
4. In the Fig 10 caption, Hz is repeated unnecessarily in two places.

Decision letter (RSOS-200514.R0)

Dear Professor Meissner,

On behalf of the Editors, I am pleased to inform you that your Manuscript RSOS-200514 entitled "Investigation of damping effects on low-frequency steady-state acoustical behaviour of coupled spaces" has been accepted for publication in Royal Society Open Science subject to minor revision in accordance with the referee suggestions. Please find the referees' comments at the end of this email.

The reviewers and handling editors have recommended publication, but also suggest some minor revisions to your manuscript. Therefore, I invite you to respond to the comments and revise your manuscript.

- Ethics statement

- Data accessibility

<http://datadryad.org/submit?journalID=RSOS&manu=RSOS-200514>

- Competing interests

- Authors' contributions

- Acknowledgements

- Funding statement

Because the schedule for publication is very tight, it is a condition of publication that you submit the revised version of your manuscript before 25-Jun-2020. Please note that the revision deadline will expire at 00.00am on this date. If you do not think you will be able to meet this date please let me know immediately.

- 1) A text file of the manuscript (tex, txt, rtf, docx or doc), references, tables (including captions) and figure captions. Do not upload a PDF as your "Main Document";
- 2) A separate electronic file of each figure (EPS or print-quality PDF preferred (either format should be produced directly from original creation package), or original software format);
- 3) Included a 100 word media summary of your paper when requested at submission. Please ensure you have entered correct contact details (email, institution and telephone) in your user account;
- 4) Included the raw data to support the claims made in your paper. You can either include your data as electronic supplementary material or upload to a repository and include the relevant doi within your manuscript. Make sure it is clear in your data accessibility statement how the data can be accessed;
- 5) All supplementary materials accompanying an accepted article will be treated as in their final form. Note that the Royal Society will neither edit nor typeset supplementary material and it will

be hosted as provided. Please ensure that the supplementary material includes the paper details where possible (authors, article title, journal name).

If your manuscript is newly submitted and subsequently accepted for publication, you will be asked to pay the article processing charge, unless you request a waiver and this is approved by Royal Society Publishing. You can find out more about the charges at <https://royalsocietypublishing.org/rsos/charges>. Should you have any queries, please contact openscience@royalsociety.org.

Best regards,

on behalf of Dr Philip Benson (Associate Editor) and R. Kerry Rowe (Subject Editor)
openscience@royalsociety.org

Reviewer comments to Author:

Reviewer: 1
Comments to the Author(s)

The authors have stated that "The way to reduce the value of the parameter D is to increase the sound attenuation inside the coupled spaces. This is due to the fact that with increased sound damping, the energy of the acoustic modes is physically attenuated, reducing the point-to-point variations in the a sound pressure level." This explanation is not correct. Because the authors' standard deviation is calculated using decibel values, changing the modal energy will not change D . The reduction in D with increasing sound absorption is due to the increase in modal bandwidth which increases the modal overlap. The increased modal overlap means that more modes are excited at a given frequency and this reduces the spatial standard deviation of the sound pressure level in the room (D). The increase of modal bandwidth with increasing sound absorption also explains why the variation of D with frequency becomes slower as the sound absorption is increased.

The authors' use of a sound source position that is equidistant (1 m) from the floor and two of walls will create an unusual situation. The authors should repeat their calculations with the distances of the source chosen so that they are not in the ratio of small integers to each other.

Reviewer: 2
Comments to the Author(s)

This is a nicely presented paper. Well done! I have four minor comments:

1. I would suggest including some practical reasons why spatial variation of pressure fields may be an important metric in the design of acoustic enclosures.
2. The authors should check their use of "specific acoustic impedance" to describe their parameter zeta. Specific acoustic impedance is defined as pressure/velocity and is given by $Z = \rho \cdot c \cdot \zeta$. Zeta is simply a convenient non-dimensional ratio of specific acoustic impedance to the characteristic impedance of the fluid.
3. There is a English usage error on line 43 of page 3.
4. In the Fig 10 caption, Hz is repeated unnecessarily in two places.

Author's Response to Decision Letter for (RSOS-200514.R0)

See Appendix A.

Decision letter (RSOS-200514.R1)

Dear Professor Meissner,

It is a pleasure to accept your manuscript entitled "Investigation of damping effects on low-frequency steady-state acoustical behaviour of coupled spaces" in its current form for publication in Royal Society Open Science. The comments of the reviewer(s) who reviewed your manuscript are included at the foot of this letter.

on behalf of Dr Philip Benson (Associate Editor) and R. Kerry Rowe (Subject Editor)
openscience@royalsociety.org

Appendix A

Responses to comments of Reviewer 1

Article title: Investigation of damping effects on low-frequency steady-state acoustical behaviour of coupled spaces

Authors: Mirosław Meissner, Krzysztof Wiśniewski

Manuscript ID: RSOS-200514

We are thankful for the reviewer's comments that helped to improve and clarify the manuscript. We hope that its revised version answers reviewer's concerns. In the following we give detailed replies to the comments (in order of occurrence). The reviewer's comments are reproduced for clarity. Changes in the manuscript corresponding to these comments were indicated by a blue color.

Comment 1

The authors have stated that "The way to reduce the value of the parameter D is to increase the sound attenuation inside the coupled spaces. This is due to the fact that with increased sound damping, the energy of the acoustic modes is physically attenuated, reducing the point-to-point variations in the a sound pressure level." This explanation is not correct. Because the authors' standard deviation is calculated using decibel values, changing the modal energy will not change D . The reduction in D with increasing sound absorption is due to the increase in modal bandwidth which increases the modal overlap. The increased modal overlap means that more modes are excited at a given frequency and this reduces the spatial standard deviation of the sound pressure level in the room (D). The increase of modal bandwidth with increasing sound absorption also explains why the variation of D with frequency becomes slower as the sound absorption is increased.

Response

Of course, the reviewer is right that this part of the paper is unclear. Therefore, in a revised version of the paper this part was replaced by:

"The way to reduce intense peaks of D is to increase the sound attenuation inside the coupled spaces. This is due to the fact that with increased sound damping, the energy of strongly localized mode is physically attenuated. Consequently, neighboring modes have a much greater impact on a distribution of a sound field for a frequency of localized mode, resulting in a reduction of point-to-point variations in a sound pressure level".

Such a behaviour of a sound field is illustrated in the figure I below. It shows distributions of the sound pressure amplitude P on the observation plane $z = 1.6$ m for the source frequency of 119.6 Hz corresponding to the frequency of strongly localized mode (figures. 5, 6 and table 1 in the paper). For walls nearly hard acoustically ($\alpha = 10^{-4}$), a distribution of the amplitude P reproduces exactly a modulus of the eigenfunction $\Phi_n(\mathbf{r})$ for this mode because for very small sound damping

$$P(\mathbf{r}) = \left\{ \left[\sum_{m=1}^{\infty} a_m \Phi_m(\mathbf{r}) \right]^2 + \left[\sum_{m=1}^{\infty} b_m \Phi_m(\mathbf{r}) \right]^2 \right\}^{1/2} \simeq (a_n^2 + b_n^2)^{1/2} |\Phi_n(\mathbf{r})|, \quad (\text{A1})$$

when $\omega = \omega_n$. If value of α increases to 10^{-3} , there is a significant decrease in P which is equivalent to reducing the sound pressure level by approximately 32 dB. This is due to energy suppression of the strongly localized mode. When mode energy drops significantly, equation (A1) is not satisfied

because the first term on the right-hand side of this equation is so small that neighboring modes are starting to contribute in creating a sound field. This is reflected in a modification of the pressure amplitude distribution [figure I(b)]. A further increase in α causes visible changes in this distribution [figure I(c)], however, for values of α changing from 0.1 to 0.3 a relative stabilization of both pressure amplitude and its distribution is observed [figure I(d), (e), (f)].

Figure I: Distribution of the sound pressure amplitude P on the observation plane $z = 1.6$ m for the source frequency of 119.6 Hz and the absorption coefficient α equal to: (a) 10^{-4} , (b) 10^{-3} , (c) 0.01, (d) 0.1, (e) 0.2, (f) 0.3.

Comment 2

The authors' use of a sound source position that is equidistant (1 m) from the floor and two of walls will create an unusual situation. The authors should repeat their calculations with the distances of the source chosen so that they are not in the ratio of small integers to each other.

Response

Calculations of the mean spatial deviation D were repeated for a sound source located at the point: (8.9 m, 4.1 m, 1.2 m), so the distances from the source to the floor and room walls are not in the ratio of small integers to each other. Calculation results are shown in figures II–V and these data are equivalent to the results in figures 5 and 7–9 in the paper, obtained for the source located at the point: (9 m, 4 m, 1 m). Since there is a significant similarity of calculation results for both source positions, similar observations can be made and similar conclusions can be drawn for the newly selected source position, as in the case of the previously chosen source position.

Figure II: Frequency dependence of the mean spatial deviation D for the absorption coefficient α of 10^{-4} . Numbered peaks occur at frequencies: 60.68 Hz, 119.6 Hz and 165.03 Hz. Sound source located at the point: (8.9 m, 4.1 m, 1.2 m).

Figure III: Frequency dependence of the mean spatial deviation D for the absorption coefficient α equal to: (a) 0.01, (b) 0.1. Sound source located at the point: (8.9 m, 4.1 m, 1.2 m).

Figure IV: Changes in D_{\max} , D_{\min} and D_{avg} with the absorption coefficient α . Colored dots indicate calculation results. Sound source located at the point: (8.9 m, 4.1 m, 1.2 m).

Figure V: Frequency dependence of the mean spatial deviation D for the absorption coefficient α equal to: (a) 0.25, (b) 0.35. Numbered peaks occur at frequencies: 37.97 Hz, 69.94 Hz, 108.7 Hz and 134.47 Hz. Sound source located at the point: (8.9 m, 4.1 m, 1.2 m).

Responses to comments of Reviewer 2

Article title: Investigation of damping effects on low-frequency steady-state acoustical behaviour of coupled spaces

Authors: Mirosław Meissner, Krzysztof Wiśniewski

Manuscript ID: RSOS-200514

We are thankful for the reviewer's comments that helped to improve and clarify the manuscript. We hope that its revised version answers reviewer's concerns. In the following we give detailed replies to the comments (in order of occurrence). The reviewer's comments are reproduced for clarity. Changes in the manuscript corresponding to these comments were indicated by a blue color.

Comment 1

I would suggest including some practical reasons why spatial variation of pressure fields may be an important metric in the design of acoustic enclosures.

Response

To emphasize practical possibilities of using the proposed method, in Section 5 the text was added:

“A spatial irregularity of a sound field occurs in small rooms because at low frequencies room acoustic quality is strongly influenced by excited room modes. This irregularity can give rise to highly position-sensitive acoustical responses that significantly limit a correct perception of speech and music. Therefore, the proposed theoretical method can be applied in the design or acoustic treatment of small rooms such as performance studios, studio control rooms, listening rooms, audio program assessment rooms and small conference and lecture rooms where speech, music or listening is part of normal use”.

Comment 2

The authors should check their use of "specific acoustic impedance" to describe their parameter zeta. Specific acoustic impedance is defined as pressure/velocity and is given by $Z = \rho \cdot c \cdot \zeta$. Zeta is simply a convenient non-dimensional ratio of specific acoustic impedance to the characteristic impedance of the fluid.

Response

According to the reviewer's comment, in a revised version of the paper the term “specific acoustic impedance” was changed to the term “normalized impedance” which has been used, for example, in the work: M. Aretz, P. Dietrich, M. Vorländer, “Application of the mirror source method for low frequency sound prediction in rectangular rooms”, *Acta Acust. Acust.*, 100(2), 306–319, 2014. We called the quantity $\zeta = Z/\rho c$ as “specific acoustic impedance” according to the nomenclature used by Kuttruff (see page 37 from H. Kuttruff, “Room acoustics”, 4th ed.), but now it seems to us that more appropriate terms are “normalized impedance” or “impedance ratio”.

Comment 3

There is a English usage error on line 43 of page 3

Response

We made changes to the text on page 3 in line 43, although we are not sure if such changes were expected by the reviewer.

Comment 4

In the Fig 10 caption, Hz is repeated unnecessarily in two places.

Response

A caption of Fig. 10 was corrected according to the reviewer's comment.

is defined by

$$Z = \left(\frac{p}{v_n} \right)_{\text{surface}} \quad (2.2)$$

where v_n denotes the velocity component normal to the wall. For non-porous walls which are excited into vibration by the sound field, the normal component of the particle velocity is identical to the velocity of the wall vibration. Like the reflection factor, the wall impedance is generally complex and a function of the angle of sound incidence.

Frequently the 'specific acoustic impedance' is used, which is the wall impedance divided by the characteristic impedance of the air:

$$\zeta = \frac{Z}{\rho_0 c} \quad (2.2a)$$

The reciprocal of the wall impedance is the 'wall admittance'; the reciprocal of ζ is called the 'specific acoustic admittance' of the wall.

As explained in Section 1.2 any complex quantity can be represented in a rectangular coordinate system (see Fig. 1.2). This holds also for the wall impedance. In this case, the length of that arrow corresponds to the magnitude of Z while its inclination angle is the phase angle of the wall impedance:

$$\mu = \arg(Z) = \arctan \left(\frac{\text{Im} Z}{\text{Re} Z} \right) \quad (2.3)$$

If the frequency changes, the impedance will usually change as well and also the length and inclination of the arrow representing it. The curve connecting the tips of all arrows is called the 'locus of the impedance in the complex plane'. A simple example of such a curve is shown in Fig. 2.9a.

2.2 Sound reflection at normal incidence

First we assume the wall to be normal to the direction in which the incident wave is travelling, which is chosen as the x -axis of a rectangular coordinate system. The wall intersects the x -axis at $x = 0$ (Fig. 2.1). The wave is coming from the left and its sound pressure is

$$p_i(x, t) = \hat{p}_0 \exp [i(\omega t - kx)] \quad (2.4a)$$

The particle velocity in the incident wave is according to eqn (1.9):

$$v_i(x, t) = \frac{\hat{p}_0}{\rho_0 c} \exp [i(\omega t - kx)] \quad (2.4b)$$